# Formal Guarantees on the Robustness of a Classifier against Adversarial Manipulation

**Matthias Hein and Maksym Andriushchenko**
Department of Mathematics and Computer Science
Saarland University, Saarbrücken Informatics Campus, Germany

## Abstract

Recent work has shown that state-of-the-art classifiers are quite brittle, in the sense that a small adversarial change of an originally with high confidence correctly classified input leads to a wrong classification again with high confidence. This raises concerns that such classifiers are vulnerable to attacks and calls into question their usage in safety-critical systems. We show in this paper for the first time formal guarantees on the robustness of a classifier by giving instance-specific *lower bounds* on the norm of the input manipulation required to change the classifier decision. Based on this analysis we propose the Cross-Lipschitz regularization functional. We show that using this form of regularization in kernel methods resp. neural networks improves the robustness of the classifier with no or small loss in prediction performance.

## 1   Introduction

The problem of adversarial manipulation of classifiers has been addressed initially in the area of spam email detection, see e.g. [5, 16]. The goal of the spammer is to manipulate the spam email (the input of the classifier) in such a way that it is not detected by the classifier. In deep learning the problem was brought up in the seminal paper by [24]. They showed for state-of-the-art deep neural networks, that one can manipulate an originally correctly classified input image with a *non-perceivable* small transformation so that the classifier now misclassifies this image with high confidence, see [7] or Figure 3 for an illustration. This property calls into question the usage of neural networks and other classifiers showing this behavior in safety critical systems, as they are vulnerable to attacks. On the other hand this also shows that the concepts learned by a classifier are still quite far away from the visual perception of humans. Subsequent research has found fast ways to generate adversarial samples with high probability [7, 12, 19] and suggested to use them during training as a form of data augmentation to gain more robustness. However, it turns out that the so-called adversarial training does not settle the problem as one can yet again construct adversarial examples for the final classifier. Interestingly, it has recently been shown that there exist universal adversarial changes which when applied lead, for every image, to a wrong classification with high probability [17]. While one needs access to the neural network model for the generation of adversarial changes, it has been shown that adversarial manipulations generalize across neural networks [18, 15, 14], which means that neural network classifiers can be attacked even as a black-box method. The most extreme case has been shown recently [15], where they attack the commercial system Clarifai, which is a black-box system as neither the underlying classifier nor the training data are known. Nevertheless, they could successfully generate adversarial images with an existing network and fool this commercial system. This emphasizes that there are indeed severe security issues with modern neural networks. While countermeasures have been proposed [8, 7, 26, 18, 12, 2], none of them provides a guarantee

of preventing this behavior [3]. One might think that generative adversarial neural networks should be resistant to this problem, but it has recently been shown [13] that they can also be attacked by adversarial manipulation of input images.

In this paper we show for the first time instance-specific formal guarantees on the robustness of a classifier against adversarial manipulation. That means we provide *lower bounds* on the norm of the change of the input required to alter the classifier decision or said otherwise: we provide a guarantee that the classifier decision does not change in a certain ball around the considered instance. We exemplify our technique for two widely used family of classifiers: kernel methods and neural networks. Based on the analysis we propose a new regularization functional, which we call *Cross-Lipschitz Regularization.* This regularization functional can be used in kernel methods and neural networks. We show that using Cross-Lipschitz regularization improves both the formal guarantees of the resulting classifier (lower bounds) as well as the change required for adversarial manipulation (upper bounds) while maintaining similar prediction performance achievable with other forms of regularization. While there exist fast ways to generate adversarial samples [7, 12, 19] without constraints, we provide algorithms based on the first order approximation of the classifier which generate adversarial samples satisfying box constraints in $O(d \log d)$, where $d$ is the input dimension.

## 2 Formal Robustness Guarantees for Classifiers

In the following we consider the multi-class setting for $K$ classes and $d$ features where one has a classifier $f : \mathbb{R}^d \to \mathbb{R}^K$ and a point $x$ is classified via $c = \arg\max_{j=1,\ldots,K} f_j(x)$. We call a classifier robust at $x$ if small changes of the input do not alter the decision. Formally, the problem can be described as follows [24]. Suppose that the classifier outputs class $c$ for input $x$, that is $f_c(x) > f_j(x)$ for $j \neq c$ (we assume the decision is unique). The problem of generating an input $x + \delta$ such that the classifier decision changes, can be formulated as

$$\min_{\delta \in \mathbb{R}^d} \|\delta\|_p, \qquad \text{s.th.} \quad \max_{l \neq c} f_l(x + \delta) \geq f_c(x + \delta) \ \text{ and } \ x + \delta \in C, \tag{1}$$

where $C$ is a constraint set specifying certain requirements on the generated input $x + \delta$, e.g., an image has to be in $[0, 1]^d$. Typically, the optimization problem (1) is non-convex and thus intractable. The so generated points $x + \delta$ are called *adversarial samples.* Depending on the $p$-norm the perturbations have different characteristics: for $p = \infty$ the perturbations are small and affect all features, whereas for $p = 1$ one gets sparse solutions up to the extreme case that only a single feature is changed. In [24] they used $p = 2$ which leads to more spread but still localized perturbations. The striking result of [24, 7] was that for most instances in computer vision datasets, the change $\delta$ necessary to alter the decision is astonishingly small and thus clearly the label should not change. However, we will see later that our new regularizer leads to robust classifiers in the sense that the required adversarial change is so large that now also the class label changes (we have found the correct decision boundary), see Fig 3. Already in [24] it is suggested to add the generated adversarial samples as a form of data augmentation during the training of neural networks in order to achieve robustness. This is denoted as *adversarial training.* Later on fast ways to approximately solve (1) were proposed in order to speed up the adversarial training process [7, 12, 19]. However, in this way, given that the approximation is successful, that is $\arg\max_{j} f_j(x + \delta) \neq c$, one gets just upper bounds on the perturbation necessary to change the classifier decision. Also it was noted early on, that the final classifier achieved by adversarial training is again vulnerable to adversarial samples [7]. Robust optimization has been suggested as a measure against adversarial manipulation [12, 21] which effectively boils down to adversarial training in practice. It is thus fair to say that up to date no mechanism exists which prevents the generation of adversarial samples nor can defend against it [3].

In this paper we focus instead on robustness guarantees, that is we show that the classifier decision does not change in a small ball around the instance. Thus our guarantees hold for any method to generate adversarial samples or input transformations due to noise or sensor failure etc. Such formal guarantees are in our point of view absolutely necessary when a classifier becomes part of a safety-critical technical system such as autonomous driving. In the following we will first show how one can achieve such a guarantee and then explicitly

derive bounds for kernel methods and neural networks. We think that such formal guarantees on robustness should be investigated further and it should become standard to report them for different classifiers alongside the usual performance measures.

## 2.1 Formal Robustness Guarantee against Adversarial Manipulation

The following guarantee holds for any classifier which is continuously differentiable with respect to the input in each output component. It is instance-specific and depends to some extent on the confidence in the decision, at least if we measure confidence by the relative difference $f_c(x) - \max_{j \neq c} f_j(x)$ as it is typical for the cross-entropy loss and other multi-class losses. In the following we use the notation $B_p(x, R) = \{y \in \mathbb{R}^d \,|\, \|x - y\|_p \leq R\}$.

**Theorem 2.1.** *Let $x \in \mathbb{R}^d$ and $f : \mathbb{R}^d \to \mathbb{R}^K$ be a multi-class classifier with continuously differentiable components and let $c = \arg\max_{j=1,\ldots,K} f_j(x)$ be the class which $f$ predicts for $x$. Let $q \in \mathbb{R}$ be defined as $\frac{1}{p} + \frac{1}{q} = 1$, then for all $\delta \in \mathbb{R}^d$ with*

$$\|\delta\|_p \leq \max_{R>0} \min \left\{ \min_{j \neq c} \frac{f_c(x) - f_j(x)}{\max_{y \in B_p(x,R)} \|\nabla f_c(y) - \nabla f_j(y)\|_q}, \; R \right\} := \alpha,$$

*it holds $c = \arg\max_{j=1,\ldots,K} f_j(x + \delta)$, that is the classifier decision does not change on $B_p(x, \alpha)$.*

Note that the bound requires in the denominator a bound on the local Lipschitz constant of all cross terms $f_c - f_j$, which we call local cross-Lipschitz constant in the following. However, we do not require to have a global bound. The problem with a global bound is that the ideal robust classifier is basically piecewise constant on larger regions with sharp transitions between the classes. However, the global Lipschitz constant would then just be influenced by the sharp transition zones and would not yield a good bound, whereas the local bound can adapt to regions where the classifier is approximately constant and then yields good guarantees. In [24, 4] they suggest to study the global Lipschitz constant[1] of each $f_j$, $j = 1, \ldots, K$. A small global Lipschitz constant for all $f_j$ implies a good bound as

$$\|\nabla f_j(y) - \nabla f_c(y)\|_q \leq \|\nabla f_j(y)\|_q + \|\nabla f_c(y)\|_q, \tag{2}$$

but the converse does not hold. As discussed below it turns out that our local estimates are significantly better than the suggested global estimates which implies also better robustness guarantees. In turn we want to emphasize that our bound is tight, that is the bound is attained, for linear classifiers $f_j(x) = \langle w_j, x \rangle$, $j = 1, \ldots, K$. It holds

$$\|\delta\|_p = \min_{j \neq c} \frac{\langle w_c - w_j, x \rangle}{\|w_c - w_j\|_q}.$$

In Section 4 we refine this result for the case when the input is constrained to $[0, 1]^d$. In general, it is possible to integrate constraints on the input by simply doing the maximum over the intersection of $B_p(x, R)$ with the constraint set e.g. $[0, 1]^d$ for gray-scale images.

## 2.2 Evaluation of the Bound for Kernel Methods

Next, we discuss how the bound can be evaluated for different classifier models. For simplicity we restrict ourselves to the case $p = 2$ (which implies $q = 2$) and leave the other cases to future work. We consider the class of kernel methods, that is the classifier has the form

$$f_j(x) = \sum_{r=1}^n \alpha_{jr} k(x_r, x),$$

where $(x_r)_{r=1}^n$ are the $n$ training points, $k : \mathbb{R}^d \times \mathbb{R}^d \to \mathbb{R}$ is a positive definite kernel function and $\alpha \in \mathbb{R}^{K \times n}$ are the trained parameters e.g. of a SVM. The goal is to upper bound the

term $\max_{y \in B_2(x,R)} \|\nabla f_j(y) - \nabla f_c(y)\|_2$ for this classifier model. A simple calculation shows

$$0 \leq \|\nabla f_j(y) - \nabla f_c(y)\|_2^2 = \sum_{r,s=1}^{n} (\alpha_{jr} - \alpha_{cr})(\alpha_{js} - \alpha_{cs}) \langle \nabla_y k(x_r, y), \nabla_y k(x_s, y) \rangle \quad (3)$$

It has been reported that kernel methods with a Gaussian kernel are robust to noise. Thus we specialize now to this class, that is $k(x,y) = e^{-\gamma \|x-y\|_2^2}$. In this case

$$\langle \nabla_y k(x_r, y), \nabla_y k(x_s, y) \rangle = 4\gamma^2 \langle y - x_r, y - x_s \rangle e^{-\gamma \|x_r - y\|_2^2} e^{-\gamma \|x_s - y\|_2^2}.$$

We derive the following bound

**Proposition 2.1.** *Let $\beta_r = \alpha_{jr} - \alpha_{cr}$, $r = 1, \ldots, n$ and define $M = \min\left\{ \frac{\|2x - x_r - x_s\|_2}{2}, R \right\}$ and $S = \|2x - x_r - x_s\|_2$. Then*

$$\max_{y \in B_2(x,R)} \|\nabla f_j(y) - \nabla f_c(y)\|_2 \leq 2\gamma$$

$$\left( \sum_{\substack{r,s=1 \\ \beta_r \beta_s \geq 0}}^{n} \beta_r \beta_s \left[ \max\{\langle x - x_r, x - x_s \rangle + RS + R^2, 0\} e^{-\gamma\left(\|x-x_r\|_2^2 + \|x-x_s\|_2^2 - 2MS + 2M^2\right)} \right. \right.$$

$$\left. + \min\{\langle x - x_r, x - x_s \rangle + RS + R^2, 0\} e^{-\gamma\left(\|x-x_r\|_2^2 + \|x-x_s\|_2^2 + 2RS + 2R^2\right)} \right]$$

$$+ \sum_{\substack{r,s=1 \\ \beta_r \beta_s < 0}}^{n} \beta_r \beta_s \left[ \max\{\langle x - x_r, x - x_s \rangle - MS + M^2, 0\} e^{-\gamma\left(\|x-x_r\|_2^2 + \|x-x_s\|_2^2 + 2RS + 2R^2\right)} \right.$$

$$\left. \left. + \min\{\langle x - x_r, x - x_s \rangle - MS + M^2, 0\} e^{-\gamma\left(\|x-x_r\|_2^2 + \|x-x_s\|_2^2 - 2MS + 2M^2\right)} \right] \right)^{\frac{1}{2}}$$

While the bound leads to non-trivial estimates as seen in Section 5, the bound is not very tight. The reason is that the sum is bounded elementwise, which is quite pessimistic. We think that better bounds are possible but have to postpone this to future work.

## 2.3 Evaluation of the Bound for Neural Networks

We derive the bound for a neural network with one hidden layer. In principle, the technique we apply below can be used for arbitrary layers but the computational complexity increases rapidly. The problem is that in the directed network topology one has to consider almost each path separately to derive the bound. Let $U$ be the number of hidden units and $w, u$ are the weight matrices of the output resp. input layer. We assume that the activation function $\sigma$ is continuously differentiable and assume that the derivative $\sigma'$ is monotonically increasing. Our prototype activation function we have in mind and which we use later on in the experiment is the differentiable approximation, $\sigma_\alpha(x) = \frac{1}{\alpha} \log(1 + e^{\alpha x})$ of the ReLU activation function $\sigma_{\text{ReLU}}(x) = \max\{0, x\}$. Note that $\lim_{\alpha \to \infty} \sigma_\alpha(x) = \sigma_{\text{ReLU}}(x)$ and $\sigma'_\alpha(x) = \frac{1}{1 + e^{-\alpha x}}$. The output of the neural network can be written as

$$f_j(x) = \sum_{r=1}^{U} w_{jr} \sigma\left( \sum_{s=1}^{d} u_{rs} x_s \right), \quad j = 1, \ldots, K,$$

where for simplicity we omit any bias terms, but it is straightforward to consider also models with bias. A direct computation shows that

$$\|\nabla f_j(y) - \nabla f_c(y)\|_2^2 = \sum_{r,m=1}^{U} (w_{jr} - w_{cr})(w_{jm} - w_{cm})\sigma'(\langle u_r, y \rangle)\sigma'(\langle u_m, y \rangle) \sum_{l=1}^{d} u_{rl} u_{ml}, \quad (4)$$

where $u_r \in \mathbb{R}^d$ is the $r$-th row of the weight matrix $u \in \mathbb{R}^{U \times d}$. The resulting bound is given in the following proposition.

**Proposition 2.2.** *Let $\sigma$ be a continuously differentiable activation function with $\sigma'$ monotonically increasing. Define $\beta_{rm} = (w_{jr} - w_{cr})(w_{jm} - w_{cm}) \sum_{l=1}^{d} u_{rl} u_{ml}$. Then*

$$\max_{y \in B_2(x,R)} \|\nabla f_j(y) - \nabla f_c(y)\|_2$$

$$\leq \Big[ \sum_{r,m=1}^{U} \max\{\beta_{rm}, 0\} \sigma'\big(\langle u_r, x\rangle + R\|u_r\|_2\big) \sigma'\big(\langle u_m, x\rangle + R\|u_m\|_2\big)$$

$$+ \min\{\beta_{rm}, 0\} \sigma'\big(\langle u_r, x\rangle - R\|u_r\|_2\big) \sigma'\big(\langle u_m, x\rangle - R\|u_m\|_2\big) \Big]^{\frac{1}{2}}$$

As discussed above the global Lipschitz bounds of the individual classifier outputs, see (2), lead to an upper bound of our desired local cross-Lipschitz constant. In the experiments below our local bounds on the Lipschitz constant are up to 8 times smaller, than what one would achieve via the global Lipschitz bounds of [24]. This shows that their global approach is much too rough to get meaningful robustness guarantees.

## 3 The Cross-Lipschitz Regularization Functional

We have seen in Section 2 that if

$$\max_{j \neq c} \max_{y \in B_p(x,R)} \|\nabla f_c(y) - \nabla f_j(y)\|_q , \tag{5}$$

is small and $f_c(x) - f_j(x)$ is large, then we get good robustness guarantees. The latter property is typically already optimized in a multi-class loss function. We consider for all methods in this paper the cross-entropy loss so that the differences in the results only come from the chosen function class (kernel methods versus neural networks) and the chosen regularization functional. The cross-entropy loss $L : \{1, \ldots, K\} \times \mathbb{R}^K \to \mathbb{R}$ is given as

$$L(y, f(x)) = -\log\Big(\frac{e^{f_y(x)}}{\sum_{k=1}^{K} e^{f_k(x)}}\Big) = \log\Big(1 + \sum_{k \neq y}^{K} e^{f_k(x) - f_y(x)}\Big).$$

In the latter formulation it becomes apparent that the loss tries to make the difference $f_y(x) - f_k(x)$ as large as possible for all $k = 1, \ldots, K$.

As our goal are good robustness guarantees it is natural to consider a proxy of the quantity in (5) for regularization. We define the **Cross-Lipschitz Regularization** functional as

$$\Omega(f) = \frac{1}{nK^2} \sum_{i=1}^{n} \sum_{l,m=1}^{K} \|\nabla f_l(x_i) - \nabla f_m(x_i)\|_2^2 , \tag{6}$$

where the $(x_i)_{i=1}^n$ are the training points. The goal of this regularization functional is to make the *differences* of the classifier functions at the data points as constant as possible. In total by minimizing

$$\frac{1}{n} \sum_{i=1}^{n} L\big(y_i, f(x_i)\big) + \lambda \Omega(f), \tag{7}$$

over some function class we thus try to maximize $f_c(x_i) - f_j(x_i)$ and at the same time keep $\|\nabla f_l(x_i) - \nabla f_m(x_i)\|_2^2$ small uniformly over all classes. This automatically enforces robustness of the resulting classifier. It is important to note that this regularization functional is coherent with the loss as it shares the same degrees of freedom, that is adding the same function $g$ to all outputs: $f'_j(x) = f_j(x) + g(x)$ leaves loss and regularization functional invariant. This is the main difference to [4], where they enforce the global Lipschitz constant to be smaller than one.

### 3.1 Cross-Lipschitz Regularization in Kernel Methods

In kernel methods one uses typically the regularization functional induced by the kernel which is given as the squared norm of the function, $f(x) = \sum_{i=1}^{n} \alpha_i k(x_i, x)$, in the corresponding

reproducing kernel Hilbert space $H_k$, $\|f\|_{H_k}^2 = \sum_{i,j=1}^n \alpha_i \alpha_j k(x_i, x_j)$. In particular, for translation invariant kernels one can make directly a connection to penalization of derivatives of the function $f$ via the Fourier transform, see [20]. However, penalizing higher-order derivatives is irrelevant for achieving robustness. Given the kernel expansion of $f$, one can write the Cross-Lipschitz regularization function as

$$\Omega(f) = \frac{1}{nK^2} \sum_{i,j=1}^n \sum_{l,m=1}^K \sum_{r,s=1}^n (\alpha_{lr} - \alpha_{mr})(\alpha_{ls} - \alpha_{ms}) \langle \nabla_y k(x_r, x_i), \nabla_y k(x_s, x_i) \rangle$$

$\Omega$ is convex in $\alpha \in \mathbb{R}^{K \times n}$ as $k'(x_r, x_s) = \langle \nabla_y k(x_r, x_i), \nabla_y k(x_s, x_i) \rangle$ is a positive definite kernel for any $x_i$ and with the convex cross-entropy loss the learning problem in (7) is convex.

## 3.2 Cross-Lipschitz Regularization in Neural Networks

The standard way to regularize neural networks is *weight decay*; that is, the squared Euclidean norm of all weights is added to the objective. More recently dropout [22], which can be seen as a form of stochastic regularization, has been introduced. Dropout can also be interpreted as a form of regularization of the weights [22, 10]. It is interesting to note that classical regularization functionals which penalize derivatives of the resulting classifier function are not typically used in deep learning, but see [6, 11]. As noted above we restrict ourselves to one hidden layer neural networks to simplify notation, that is, $f_j(x) = \sum_{r=1}^U w_{jr} \, \sigma\big(\sum_{s=1}^d u_{rs} x_s\big), \quad j = 1, \ldots, K$. Then we can write the Cross-Lipschitz regularization as

$$\Omega(f) = \frac{2}{nK^2} \sum_{r,s=1}^U \Big( \sum_{l=1}^K w_{lr} w_{ls} - \sum_{l=1}^K w_{lr} \sum_{m=1}^K w_{ms} \Big) \sum_{i,j=1}^n \sigma'(\langle u_r, x_i \rangle) \sigma'(\langle u_s, x_i \rangle) \sum_{l=1}^d u_{rl} u_{sl}$$

which leads to an expression which can be fast evaluated using vectorization. Obviously, one can also implement the Cross-Lipschitz Regularization also for all standard deep networks.

## 4 Box Constrained Adversarial Sample Generation

The main emphasis of this paper are robustness guarantees without resorting to particular ways how to generate adversarial samples. On the other hand while Theorem 2.1 gives lower bounds on the required input transformation, efficient ways to approximately solve the adversarial sample generation in (1) are helpful to get upper bounds on the required change. Upper bounds allow us to check how tight our derived lower bounds are. As all of our experiments will be concerned with images, it is reasonable that our adversarial samples are also images. However, up to our knowledge, the current main techniques to generate adversarial samples [7, 12, 19] integrate box constraints by clipping the results to $[0,1]^d$. We provide in the following fast algorithms to generate adversarial samples which lie in $[0,1]^d$. The strategy is similar to [12], where they use a linear approximation of the classifier to derive adversarial samples with respect to different norms. Formally,

$$f_j(x + \delta) \approx f_j(x) + \langle \nabla f_j(x), \delta \rangle, \quad j = 1, \ldots, K.$$

Assuming that the linear approximation holds, the optimization problem (1) integrating box constraints for changing class $c$ into $j$ becomes

$$\min_{\delta \in \mathbb{R}^d} \ \|\delta\|_p \tag{8}$$
$$\text{sbj. to: } f_j(x) - f_c(x) \geq \langle \nabla f_c(x) - \nabla f_j(x), \delta \rangle$$
$$0 \leq x_j + \delta_j \leq 1$$

In order to get the minimal adversarial sample we have to solve this for all $j \neq c$ and take the one with minimal $\|\delta\|_p$. This yields the minimal adversarial change for linear classiifers. Note that (8) is a convex optimization problem, which can be reduced to a one-parameter problem in the dual. This allows to derive the following result (proofs and algorithms are in the supplement).

**Proposition 4.1.** *Let $p \in \{1, 2, \infty\}$, then (8) can be solved in $O(d \log d)$ time.*

For nonlinear classifiers a change of the decision is not guaranteed and thus we use later on a binary search with a variable $c$ instead of $f_c(x) - f_j(x)$.

## 5 Experiments

The goal of the experiments is the evaluation of the robustness of the resulting classifiers and not necessarily state-of-the-art results in terms of test error. In all cases we compute the robustness guarantees from Theorem 2.1 (lower bound on the norm of the minimal change required to change the classifier decision), where we optimize over $R$ using binary search, and adversarial samples with the algorithm for the 2-norm from Section 4 (upper bound on the norm of the minimal change required to change the classifier decision), where we do a binary search in the classifier output difference in order to find a point on the decision boundary. Additional experiments can be found in the supplementary material.

**Kernel methods:** We optimize the cross-entropy loss once with the standard regularization (Kernel-LogReg) and with Cross-Lipschitz regularization (Kernel-CL). Both are convex optimization problems and we use L-BFGS to solve them. We use the Gaussian kernel $k(x,y) = e^{-\gamma\|x-y\|^2}$ where $\gamma = \frac{\alpha}{\rho_{\mathrm{KNN40}}^2}$ and $\rho_{\mathrm{KNN40}}$ is the mean of the 40 nearest neighbor distances on the training set and $\alpha \in \{0.5, 1, 2, 4\}$. We show the results for MNIST (60000 training and 10000 test samples). However, we have checked that parameter selection using a subset of 50000 images from the training set and evaluating on the rest yields indeed the parameters which give the best test errors when trained on the full set. The regularization parameter is chosen in $\lambda \in \{10^{-k}|k \in \{5, 6, 7, 8\}\}$ for Kernel-SVM and $\lambda \in \{10^{-k} \,|\, k \in \{0, 1, 2, 3\}\}$ for our Kernel-CL. The results of the optimal parameters are given in the following table and the performance of all parameters is shown in Figure 1. Note that due to the high computational complexity we could evaluate the robustness guarantees only for the optimal parameters.

|  | test error | avg. $\|\cdot\|_2$ adv. samples | avg. $\|\cdot\|_2$ rob. guar. |
|---|---|---|---|
| No Reg. ($\lambda = 0$) | 2.23% | 2.39 | 0.037 |
| K-SVM | 1.48% | 1.91 | 0.058 |
| K-CL | 1.44% | 3.12 | 0.045 |

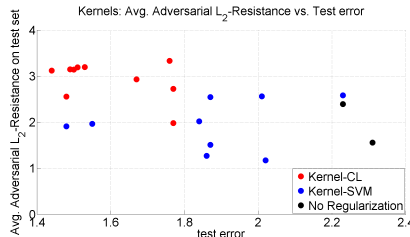

**Figure 1: Kernel Methods:** Cross-Lipschitz regularization achieves both better test error and robustness against adversarial samples (upper bounds, larger is better) compared to the standard regularization. The robustness guarantee is weaker than for neural networks but this is most likely due to the relatively loose bound.

**Neural Networks:** Before we demonstrate how upper and lower bounds improve using cross-Lipschitz regularization, we first want to highlight the importance of the usage of the local cross-Lipschitz constant in Theorem 2.1 for our robustness guarantee.

**Local versus global Cross-Lipschitz constant:** While no robustness guarantee has been proven before, it has been discussed in [24] that penalization of the global Lipschitz constant should improve robustness, see also [4]. For that purpose they derive the Lipschitz constants of several different layers and use the fact that the Lipschitz constant of a composition of functions is upper bounded by the product of the Lipschitz constants of the functions. In analogy, this would mean that the term $\sup_{y \in B(x,R)} \|\nabla f_c(y) - \nabla f_j(y)\|_2$, which we have upper bounded in Proposition 2.2, in the denominator in Theorem 2.1 could be replaced[2] by the global Lipschitz constant of $g(x) := f_c(x) - f_j(x)$. which is given as $\sup_{y \in \mathbb{R}^d} \|\nabla g(x)\|_2 = \sup_{x \neq y} \frac{|g(x) - g(y)|}{\|x-y\|_2}$. We have with $\|U\|_{2,2}$ being the largest singular value of $U$,

$$|g(x) - g(y)| = \langle w_c - w_j, \sigma(Ux) - \sigma(Uy)\rangle \leq \|w_c - w_j\|_2 \|\sigma(Ux) - \sigma(Uy)\|_2$$
$$\leq \|w_c - w_j\|_2 \|U(x-y)\|_2 \leq \|w_c - w_j\|_2 \|U\|_{2,2} \|x-y\|_2,$$

where we used that $\sigma$ is contractive as $\sigma'(z) = \frac{1}{1+e^{-\alpha z}}$ and thus we get

$$\sup_{y \in \mathbb{R}^d} \|\nabla f_c(x) - \nabla f_j(x)\|_2 \leq \|w_c - w_j\|_2 \|U\|_{2,2}.$$

| MNIST (plain) | | | | CIFAR10 (plain) | | | |
|---|---|---|---|---|---|---|---|
| None | Dropout | Weight Dec. | Cross Lip. | None | Dropout | Weight Dec. | Cross Lip. |
| 0.69 | 0.48 | 0.68 | 0.21 | 0.22 | 0.13 | 0.24 | 0.17 |

**Table 1:** We show the average ratio $\frac{\alpha_{\text{global}}}{\alpha_{\text{local}}}$ of the robustness guarantees $\alpha_{\text{global}}, \alpha_{\text{local}}$ from Theorem 2.1 on the test data for MNIST and CIFAR10 and different regularizers. The guarantees using the local Cross-Lipschitz constant are up to eight times better than with the global one.

The advantage is clearly that this global Cross-Lipschitz constant can just be computed once and by using it in Theorem 2.1 one can evaluate the guarantees very quickly. However, it turns out that one gets significantly better robustness guarantees by using the local Cross-Lipschitz constant in terms of the bound derived in Proposition 2.2 instead of the just derived global Lipschitz constant. Note that the optimization over $R$ in Theorem 2.1 is done using a binary search, noting that the bound of the local Lipschitz constant in Proposition 2.2 is monotonically decreasing in $R$. We have the following comparison in Table 1. We want to highlight that the robustness guarantee with the global Cross-Lipschitz constant was *always* worse than when using the local Cross-Lipschitz constant across all regularizers and data sets. Table 1 shows that the guarantees using the local Cross-Lipschitz can be up to eight times better than for the global one. As these are just one hidden layer networks, it is obvious that robustness guarantees for deep neural networks based on the global Lipschitz constants will be too coarse to be useful.

**Experiments:** We use a one hidden layer network with 1024 hidden units and the softplus activation function with $\alpha = 10$. Thus the resulting classifier is continuously differentiable. We compare three different regularization techniques: weight decay, dropout and our Cross-Lipschitz regularization. Training is done with SGD. For each method we have adapted the learning rate (two per method) and regularization parameters (4 per method) so that all methods achieve good performance. We do experiments for MNIST and CIFAR10 in three settings: plain, data augmentation and adversarial training. The exact settings of the parameters and the augmentation techniques are described in the supplementary material.The results for MNIST are shown in Figure 2 and the results for CIFAR10 are in the supplementary material.For MNIST there is a clear trend that our Cross-Lipschitz regularization improves the robustness of the resulting classifier while having competitive resp. better test error. It is surprising that data augmentation does not lead to more robust models. However, adversarial training improves the guarantees as well as adversarial resistance. For CIFAR10 the picture is mixed, our CL-Regularization performs well for the augmented task in test error and upper bounds but is not significantly better in the robustness guarantees. The problem might be that the overall bad performance due to the simple model is preventing a better behavior. Data augmentation leads to better test error but the robustness properties (upper and lower bounds) are basically unchanged. Adversarial training slightly improves performance compared to the plain setting and improves upper and lower bounds in terms of robustness. We want to highlight that our guarantees (lower bounds) and the upper bounds from the adversarial samples are not too far away.

**Illustration of adversarial samples:** we take one test image from MNIST and apply the adversarial generation from Section 4 wrt to the 2-norm to generate the adversarial samples for the different kernel methods and neural networks (plain setting), where we use for each method the parameters leading to best test performance. All classifiers change their originally correct decision to a "wrong" one. It is interesting to note that for Cross-Lipschitz regularization (both kernel method and neural network) the "adversarial" sample is really at the decision boundary between 1 and 8 (as predicted) and thus the new decision is actually correct. This effect is strongest for our Kernel-CL, which also requires the strongest modification to generate the adversarial sample. The situation is different for neural networks, where the classifiers obtained from the two standard regularization techniques are still vulnerable, as the adversarial sample is still clearly a 1 for dropout and weight decay.

**Outlook** Formal guarantees on machine learning systems are becoming increasingly more important as they are used in safety-critical systems. We think that there should be more

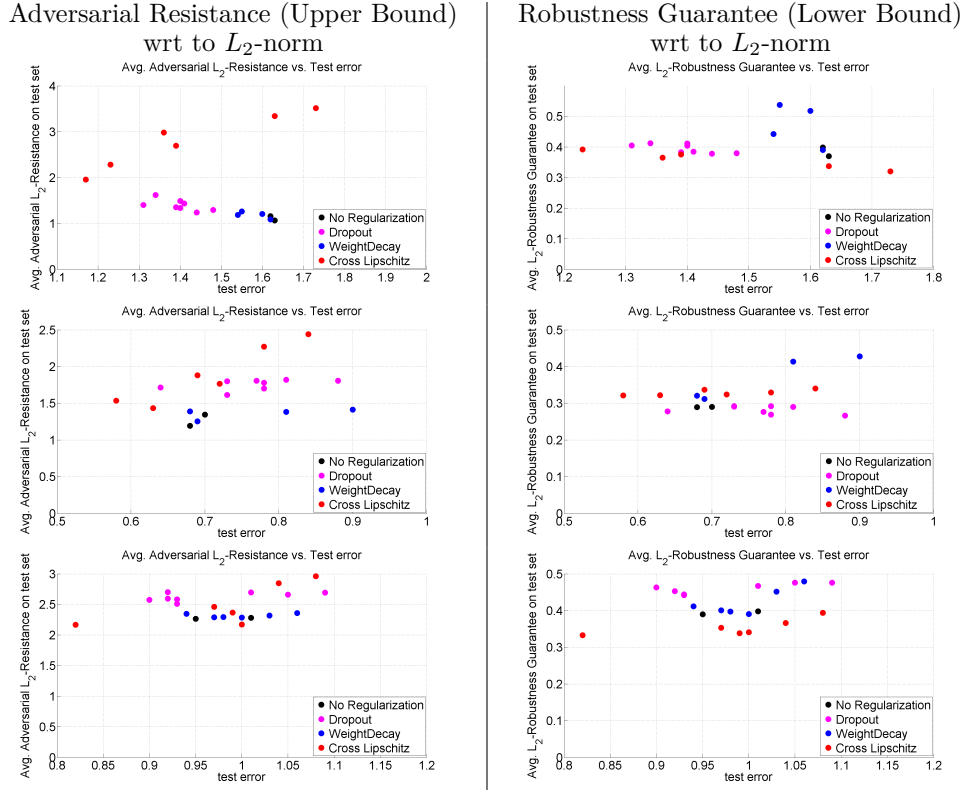

**Figure 2:** **Neural Networks, Left:** Adversarial resistance wrt to $L_2$-norm on MNIST. **Right:** Average robustness guarantee wrt to $L_2$-norm on MNIST for different neural networks (one hidden layer, 1024 HU) and hyperparameters. The Cross-Lipschitz regularization leads to better robustness with similar or better prediction performance. **Top row:** plain MNIST, **Middle:** Data Augmentation, **Bottom:** Adv. Training

research on robustness guarantees (lower bounds), whereas current research is focused on new attacks (upper bounds). We have argued that our instance-specific guarantees using our local Cross-Lipschitz constant is more effective than using a global one and leads to lower bounds which are up to 8 times better. A major open problem is to come up with tight lower bounds for deep networks.

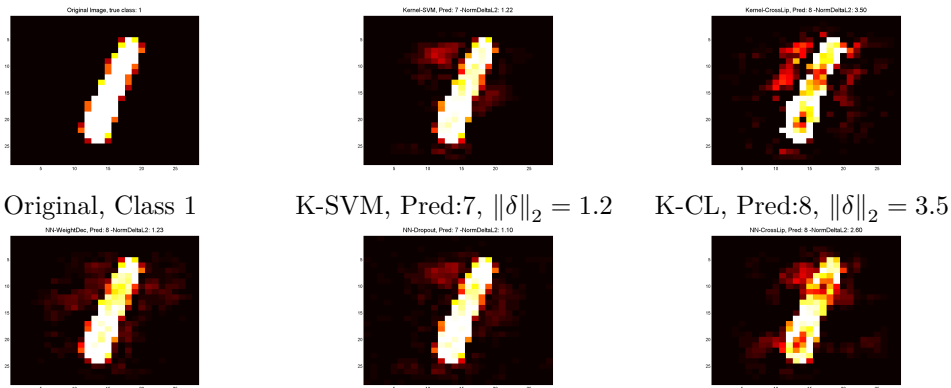

Original, Class 1      K-SVM, Pred:7, $\|\delta\|_2 = 1.2$      K-CL, Pred:8, $\|\delta\|_2 = 3.5$

NN-WD, Pred:8, $\|\delta\|_2 = 1.2$      NN-DO, Pred:7, $\|\delta\|_2 = 1.1$      NN-CL, Pred:8, $\|\delta\|_2 = 2.6$

**Figure 3:** Top left: original test image, for each classifier we generate the corresponding adversarial sample which changes the classifier decision (denoted as Pred). Note that for Cross-Lipschitz regularization this new decision makes (often) sense, whereas for the neural network models (weight decay/dropout) the change is so small that the new decision is clearly wrong.

## Footnotes

[1]The Lipschitz constant $L$ wrt to $p$-norm of a piecewise continuously differentiable function is given as $L = \sup_{x \in \mathbb{R}^d} \|\nabla f(x)\|_q$. Then it holds, $|f(x) - f(y)| \leq L \|x - y\|_p$.

[2]Note that then the optimization of $R$ in Theorem 2.1 would be unnecessary.

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
