[Supplementary Material]

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

*Proof.* By the main theorem of calculus, it holds that

$$f_j(x + \delta) = f_j(x) + \int_0^1 \langle \nabla f_j(x + t\delta), \delta \rangle \, dt, \qquad \text{for } j = 1, \ldots, K.$$

Thus, in order to achieve $f_j(x + \delta) \geq f_c(x + \delta)$, it has to hold that

$$0 \leq f_c(x) - f_j(x) \leq \int_0^1 \langle \nabla f_j(x + t\delta) - \nabla f_c(x + t\delta), \delta \rangle \, dt$$

$$\leq \|\delta\|_p \int_0^1 \|\nabla f_j(x + t\delta) - \nabla f_c(x + t\delta)\|_q \, dt,$$

where the first inequality holds as $f_c(x) \geq f_j(x)$ for all $j = 1, \ldots, K$ and in the last step we have used Hölder inequality together with the fact that the $q$-norm is dual to the $p$-norm, where $q$ is defined via $\frac{1}{p} + \frac{1}{q} = 1$. Thus the minimal norm of the change $\delta$ required to change the classifier decision from $c$ to $j$ satisfies

$$\|\delta\|_p \geq \frac{f_c(x) - f_j(x)}{\int_0^1 \|\nabla f_j(x + t\delta) - \nabla f_c(x + t\delta)\|_q \, dt}.$$

We upper bound the denominator over some fixed ball $B_p(x, R)$. Note that by doing this, we can only make assertions of perturbations $\delta \in B_p(0, R)$ and thus the upper bound in the guarantee is at most $R$. It holds

$$\sup_{\delta \in B_p(0,R)} \int_0^1 \|\nabla f_j(x + t\delta) - \nabla f_c(x + t\delta)\|_q \, dt \;\leq\; \max_{y \in B_p(x,R)} \|\nabla f_j(y) - \nabla f_c(y)\|_q.$$

Thus we get the lower bound for the minimal norm of the change $\delta$ required to change the classifier decision from $c$ to $j$,

$$\|\delta\|_p \geq \min \left\{ R, \frac{f_c(x) - f_j(x)}{\max_{y \in B_p(x,R)} \|\nabla f_j(y) - \nabla f_c(y)\|_q} \right\} := \alpha.$$

As we are interested in the worst case, we take the minimum over all $j \neq c$. Finally, the result holds for any fixed $R > 0$ so that we can maximize over $R$ which yields the final result. □

Note that the bound requires in the denominator a bound on the local Lipschitz constant of all cross terms $f_c - f_j$, which we call local cross-Lipschitz constant in the following. However, we do not require to have a global bound. The problem with a global bound is that the ideal robust classifier is basically piecewise constant on larger regions with sharp transitions between the classes. However, the global Lipschitz constant would then just be influenced by the sharp transition zones and would not yield a good bound, whereas the local bound can adapt to regions where the classifier is approximately constant and then yields good guarantees. In [24, 4] they suggest to study the global Lipschitz constant[1] of each $f_j$, $j = 1, \ldots, K$. A small global Lipschitz constant for all $f_j$ implies a good bound as

$$\|\nabla f_j(y) - \nabla f_c(y)\|_q \leq \|\nabla f_j(y)\|_q + \|\nabla f_c(y)\|_q, \tag{2}$$

but the converse does not hold. As discussed below it turns out that our local estimates are significantly better than the suggested global estimates which implies also better robustness guarantees. In turn we want to emphasize that our bound is tight, that is the bound is attained, for linear classifiers $f_j(x) = \langle w_j, x \rangle$, $j = 1, \ldots, K$. It holds

$$\|\delta\|_p = \min_{j \neq c} \frac{\langle w_c - w_j, x \rangle}{\|w_c - w_j\|_q}.$$

In Section 4 we refine this result for the case when the input is constrained to $[0,1]^d$. In general, it is possible to integrate constraints on the input by simply doing the maximum over the intersection of $B_p(x, R)$ with the constraint set e.g. $[0,1]^d$ for gray-scale images.

## 2.2 Evaluation of the Bound for Kernel Methods

Next, we discuss how the bound can be evaluated for different classifier models. For simplicity we restrict ourselves to the case $p = 2$ (which implies $q = 2$) and leave the other cases to future work. We consider the class of kernel methods, that is the classifier has the form

$$f_j(x) = \sum_{r=1}^{n} \alpha_{jr} k(x_r, x),$$

where $(x_r)_{r=1}^n$ are the $n$ training points, $k : \mathbb{R}^d \times \mathbb{R}^d \to \mathbb{R}$ is a positive definite kernel function and $\alpha \in \mathbb{R}^{K \times n}$ are the trained parameters e.g. of a SVM. The goal is to upper bound the term $\max_{y \in B_2(x,R)} \|\nabla f_j(y) - \nabla f_c(y)\|_2$ for this classifier model. A simple calculation shows

$$0 \leq \|\nabla f_j(y) - \nabla f_c(y)\|_2^2 = \sum_{r,s=1}^{n} (\alpha_{jr} - \alpha_{cr})(\alpha_{js} - \alpha_{cs}) \langle \nabla_y k(x_r, y), \nabla_y k(x_s, y) \rangle \tag{3}$$

It has been reported that kernel methods with a Gaussian kernel are robust to noise. Thus we specialize now to this class, that is $k(x, y) = e^{-\gamma \|x-y\|_2^2}$. In this case

$$\langle \nabla_y k(x_r, y), \nabla_y k(x_s, y) \rangle = 4\gamma^2 \langle y - x_r, y - x_s \rangle e^{-\gamma \|x_r - y\|_2^2} e^{-\gamma \|x_s - y\|_2^2}.$$

We will now derive lower and upper bounds on this term uniformly over $B_2(x, R)$ which allows us to derive the guarantee.

**Lemma 2.1.** *Let* $M = \min \left\{ \frac{\|2x - x_r - x_s\|_2}{2}, R \right\}$, *then*

$$\max_{y \in B_2(x,R)} \langle y - x_r, y - x_s \rangle = \langle x - x_r, x - x_s \rangle + R \|2x - x_r - x_s\|_2 + R^2$$

$$\min_{y \in B_2(x,R)} \langle y - x_r, y - x_s \rangle = \langle x - x_r, x - x_s \rangle - M \|2x - x_r - x_s\|_2 + M^2$$

$$\max_{y \in B_2(x,R)} e^{-\gamma \|x_r - y\|_2^2} e^{-\gamma \|x_s - y\|_2^2} = e^{-\gamma \left( \|x - x_r\|_2^2 + \|x - x_s\|_2^2 - 2M \|2x - x_r - x_s\|_2 + 2M^2 \right)}$$

$$\min_{y \in B_2(x,R)} e^{-\gamma \|x_r - y\|_2^2} e^{-\gamma \|x_s - y\|_2^2} = e^{-\gamma \left( \|x - x_r\|_2^2 + \|x - x_s\|_2^2 + 2R \|2x - x_r - x_s\|_2 + 2R^2 \right)}$$

*Proof.* For the first part we use

$$\max_{y \in B_2(x,R)} \langle y - x_r, y - x_s \rangle = \max_{h \in B_2(0,R)} \langle x - x_r + h, x - x_s + h \rangle$$

$$= \langle x - x_r, x - x_s \rangle + \max_{h \in B_2(0,R)} \langle h, 2x - x_r - x_s \rangle + \|h\|_2^2$$

$$= \langle x - x_r, x - x_s \rangle + R \|2x - x_r - x_s\|_2 + R^2,$$

where the last equality follows by Cauchy-Schwarz and noting that equality is attained as we maximize over the Euclidean unit ball. For the second part we consider

$$\min_{y \in B_2(x,R)} \langle y - x_r, y - x_s \rangle = \min_{h \in B_2(0,R)} \langle x - x_r + h, x - x_s + h \rangle$$

$$= \langle x - x_r, x - x_s \rangle + \min_{h \in B_2(0,R)} \langle h, 2x - x_r - x_s \rangle + \|h\|_2^2$$

$$= \langle x - x_r, x - x_s \rangle + \min_{0 \le \alpha \le R} -\alpha \|2x - x_r - x_s\|_2 + \alpha^2$$

$$= \langle x - x_r, x - x_s \rangle - \min \left\{ \frac{\|2x - x_r - x_s\|_2}{2}, R \right\} \|2x - x_r - x_s\|_2$$

$$+ \left( \min \left\{ \frac{\|2x - x_r - x_s\|_2}{2}, R \right\} \right)^2,$$

where in the second step we have separated direction and norm of the vector, optimization over the direction yields with Cauchy-Schwarz the result. Finally, the constrained convex one-dimensional optimization problem can be solved explicitly as $\alpha = \min\{\frac{\|2x - x_r - x_s\|_2}{2}, R\}$. The proof of the other results follows analogously noting that

$$e^{-\gamma \|x + h - x_r\|_2^2} e^{-\gamma \|x + h - x_s\|_2^2} = e^{-\gamma \left( \|x - x_r\|_2^2 + \|x - x_s\|_2^2 + 2\langle h, 2x - x_r - x_s \rangle + 2\|h\|_2^2 \right)}.$$

$$\square$$

Using this lemma it is easy to derive the final result.

**Proposition 2.1.** *Let $\beta_r = \alpha_{jr} - \alpha_{cr}$, $r = 1, \ldots, n$ and define $M = \min \left\{ \frac{\|2x - x_r - x_s\|_2}{2}, R \right\}$ and $S = \|2x - x_r - x_s\|_2$. Then*

$$\max_{y \in B_2(x,R)} \|\nabla f_j(y) - \nabla f_c(y)\|_2 \le 2\gamma$$

$$\left( \sum_{\substack{r,s=1 \\ \beta_r \beta_s \ge 0}}^{n} \beta_r \beta_s \left[ \max\{\langle x - x_r, x - x_s \rangle + RS + R^2, 0\} e^{-\gamma \left( \|x - x_r\|_2^2 + \|x - x_s\|_2^2 - 2MS + 2M^2 \right)} \right. \right.$$

$$\left. + \min\{\langle x - x_r, x - x_s \rangle + RS + R^2, 0\} e^{-\gamma \left( \|x - x_r\|_2^2 + \|x - x_s\|_2^2 + 2RS + 2R^2 \right)} \right]$$

$$+ \sum_{\substack{r,s=1 \\ \beta_r \beta_s < 0}}^{n} \beta_r \beta_s \left[ \max\{\langle x - x_r, x - x_s \rangle - MS + M^2, 0\} e^{-\gamma \left( \|x - x_r\|_2^2 + \|x - x_s\|_2^2 + 2RS + 2R^2 \right)} \right.$$

$$\left. \left. + \min\{\langle x - x_r, x - x_s \rangle - MS + M^2, 0\} e^{-\gamma \left( \|x - x_r\|_2^2 + \|x - x_s\|_2^2 - 2MS + 2M^2 \right)} \right] \right)^{\frac{1}{2}}$$

*Proof.* We bound each term in the sum in Equation 3 separately using that $ac \le bd$ if $b, c \ge 0$, $a \le b$ and $c \le d$ or $b \le 0$, $d \ge 0$, $a \le b$ and $c \ge d$, where $c, d$ correspond to the exponential terms and $a, b$ to the upper bounds of the inner product. Similarly, $ab \ge cd$ if $b, c \ge 0$, $a \ge b$ and $c \ge d$ or $a \le 0$, $d \ge 0$, $a \ge b$ and $c \le d$. The individual upper and lower bounds are taken from Lemma 2.1. $\square$

While the bound leads to non-trivial estimates as seen in Section 5, the bound is not very tight. The reason is that the sum is bounded elementwise, which is quite pessimistic. We think that better bounds are possible but have to postpone this to future work.

## 2.3 Evaluation of the Bound for Neural Networks

We derive the bound for a neural network with one hidden layer. In principle, the technique we apply below can be used for arbitrary layers but the computational complexity increases rapidly. The problem is that in the directed network topology one has to consider almost each path separately to derive the bound. Let $U$ be the number of hidden units and $w, u$ are the weight matrices of the output resp. input layer. We assume that the activation function $\sigma$ is continuously differentiable and assume that the derivative $\sigma'$ is monotonically increasing. Our prototype activation function we have in mind and which we use later on in the experiment is the differentiable approximation, $\sigma_\alpha(x) = \frac{1}{\alpha} \log(1 + e^{\alpha x})$ of the ReLU activation function $\sigma_{\mathrm{ReLU}}(x) = \max\{0, x\}$. Note that $\lim_{\alpha \to \infty} \sigma_\alpha(x) = \sigma_{\mathrm{ReLU}}(x)$ and $\sigma'_\alpha(x) = \frac{1}{1 + e^{-\alpha x}}$. The output of the neural network can be written as

$$ f_j(x) = \sum_{r=1}^{U} w_{jr}\, \sigma\Big( \sum_{s=1}^{d} u_{rs} x_s \Big), \quad j = 1, \ldots, K, $$

where for simplicity we omit any bias terms, but it is straightforward to consider also models with bias. A direct computation shows that

$$ \|\nabla f_j(y) - \nabla f_c(y)\|_2^2 = \sum_{r,m=1}^{U} (w_{jr} - w_{cr})(w_{jm} - w_{cm}) \sigma'(\langle u_r, y\rangle) \sigma'(\langle u_m, y\rangle) \sum_{l=1}^{d} u_{rl} u_{ml}, \quad (4) $$

where $u_r \in \mathbb{R}^d$ is the $r$-th row of the weight matrix $u \in \mathbb{R}^{U \times d}$. The resulting bound is given in the following proposition.

**Proposition 2.2.** *Let $\sigma$ be a continuously differentiable activation function with $\sigma'$ monotonically increasing. Define $\beta_{rm} = (w_{jr} - w_{cr})(w_{jm} - w_{cm}) \sum_{l=1}^{d} u_{rl} u_{ml}$. Then*

$$ \max_{y \in B_2(x,R)} \|\nabla f_j(y) - \nabla f_c(y)\|_2 $$

$$ \leq \Big[ \sum_{r,m=1}^{U} \max\{\beta_{rm}, 0\} \sigma'\big( \langle u_r, x\rangle + R\,\|u_r\|_2 \big) \sigma'\big( \langle u_m, x\rangle + R\,\|u_m\|_2 \big) $$

$$ + \min\{\beta_{rm}, 0\} \sigma'\big( \langle u_r, x\rangle - R\,\|u_r\|_2 \big) \sigma'\big( \langle u_m, x\rangle - R\,\|u_m\|_2 \big) \Big]^{\frac{1}{2}} $$

*Proof.* The proof is based on the fact that due the monotonicity of $\sigma'$ and with Cauchy-Schwarz,

$$ \max_{y \in B_2(x,R)} \sigma'(\langle u_r, y\rangle) = \max_{h \in B_2(0,R)} \sigma'(\langle u_r, x\rangle + \langle u_r, h\rangle) = \sigma'(\langle u_r, x\rangle + R\,\|u_r\|_2). $$

Similarly, one gets

$$ \min_{y \in B_2(x,R)} \sigma'(\langle u_r, y\rangle) = \min_{h \in B_2(0,R)} \sigma'(\langle u_r, x\rangle + \langle u_r, h\rangle) = \sigma'(\langle u_r, x\rangle - R\,\|u_r\|_2). $$

The rest of the result follows by element-wise bounding the terms in the sum in Equation 4. $\square$

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

**Lemma 4.1.** *Let $m = \arg\max_j f_j(x)$ and define $v = \nabla f_m(x) - \nabla f_j(x)$ and $0 > c = f_j(x) - f_m(x)$. If a solution of problem (9) exists, then it is given as*

$$\delta_r = \begin{cases} -\lambda v_r, & if \ -x_r \leq -\lambda v_r \leq 1 - x_r, \\ 1 - x_r, & if \ -\lambda v_r \geq 1 - x_r, \\ -x_r, & if \ -x_r \geq -\lambda v_r. \end{cases}$$

*The optimal $\lambda \geq 0$ can be obtained by solving*

$$c = \langle v, \delta \rangle = -\lambda \sum_{-x_r \leq -\lambda v_r \leq 1 - x_r} v_r^2 + \sum_{-\lambda v_r > 1 - x_r} v_r(1 - x_r) - \sum_{-\lambda v_r < -x_r} v_r x_r.$$

*If Problem 9 is infeasible, then this equation has no solution for $\lambda \geq 0$. In both the feasible and infeasible case the solution can be found in $O(d \log d)$. The algorithm is given in Algorithm 1.*

*Proof.* The Lagrangian is given by

$$L(\delta, \lambda, \alpha, \beta) = \frac{1}{2} \|\delta\|_2^2 + \lambda\big(\langle v, \delta \rangle - c\big) + \langle \alpha, x + \delta - \mathbf{1} \rangle - \langle \beta, x + \delta \rangle.$$

The KKT conditions become

$$\delta + \lambda v + \alpha - \beta = 0$$
$$\alpha_r(x_r + \delta_r - 1) = 0, \quad \forall r = 1, \dots, d$$
$$\beta_r(x_r + \delta_r) = 0, \quad \forall r = 1, \dots, d$$
$$\lambda(\langle v, \delta \rangle - c) = 0$$
$$\alpha_r \geq 0, \quad \forall r = 1, \dots, d$$
$$\beta_r \geq 0, \quad \forall r = 1, \dots, d$$
$$\lambda \geq 0.$$

We deduce that if $\beta_r > 0$ then $\alpha_r = 0$ which implies

$$\delta_r = -x_r = -\lambda v_r + \beta_r \quad \Longrightarrow \quad \beta_r = \max\{0, -x_r + \lambda v_r\}.$$

Similarly, if $\alpha_r > 0$ then $\beta_r = 0$ which implies

$$\delta_r = 1 - x_r = -\lambda v_r - \alpha_r \quad \Longrightarrow \quad \alpha_r = \max\{0, x_r - 1 - \lambda v_r\}.$$

It follows

$$\delta_r = \begin{cases} -\lambda v_r & \text{if } -x_r < -\lambda v_r < 1 - x_r \\ 1 - x_r & \text{if } -\lambda v_r > 1 - x_r \\ -x_r & \text{if } -\lambda v_r < -x_r \end{cases}.$$

We can determine $\lambda$ by inspecting $\langle v, \delta \rangle$ which is given as

$$\langle v, \delta \rangle = -\lambda \sum_{-x_r \leq -\lambda v_r \leq 1 - x_r} v_r^2 + \sum_{-\lambda v_r > 1 - x_r} v_r(1 - x_r) - \sum_{-\lambda v_r < -x_r} v_r x_r.$$

Note that $\lambda \geq 0$ and $1 - x_r \geq 0$ and thus $-\lambda v_r > 1 - x_r$ implies $v_r < 0$, thus $v_r(1 - x_r) \leq 0$ and similarly $-\lambda v_r < -x_r$ implies $v_r > 0$ and thus also $-v_r x_r < 0$. Note that the term $\langle v, \delta \rangle$ is monotonically decreasing as $\lambda$ is increasing. Thus one can sort $\max\{\frac{x_r - 1}{v_r}, \frac{x_r}{v_r}\}$ in increasing order and they represent the thresholds when the summation changes. Then we compute $\langle v, \delta \rangle$ for all of these thresholds and determine the largest threshold $\lambda^*$ such that $\langle v, \delta \rangle \leq c$. This fixes the index sets of all sums. Then we determine $\lambda$ by

$$\lambda = \frac{\sum\limits_{-\lambda^* v_r > 1 - x_r} v_r(1 - x_r) - \sum\limits_{-\lambda^* v_r < -x_r} v_r x_r - c}{\sum\limits_{-x_r \leq -\lambda^* v_r \leq 1 - x_r} v_r^2}.$$

In total sorting takes time $O(d \log d)$ and solving for $\lambda^*$ has complexity $O(d)$. $\qquad\square$

Next we consider the case $p = 1$.

$$\min_{\delta \in \mathbb{R}^d} \|\delta\|_1 \tag{10}$$
$$\text{sbj. to: } f_j(x) - f_c(x) \geq \langle \nabla f_c(x) - \nabla f_j(x), \delta \rangle$$
$$0 \leq x_j + \delta_j \leq 1$$

**Lemma 4.2.** *Let $m = \arg\max\limits_{j} f_j(x)$ and define $v = \nabla f_m(x) - \nabla f_j(x)$ and $c = f_j(x) - f_m(x) < 0$, then the solution of the problem* (10) *can be found by Algorithm 2.*

*Proof.* The result is basically obvious but we derive it formally. First of all we rewrite (10) as a linear program.

$$\min_{\delta \in \mathbb{R}^d} \sum_{i=1}^d t_i \tag{11}$$
$$\text{sbj. to: } c \geq \langle v, \delta \rangle$$
$$-x_j \leq \delta_j \leq 1 - x_j$$
$$-t_i \leq \delta_i \leq t_i$$
$$t_i \geq 0$$

**Algorithm 1** Computation of box-constrained adversarial samples wrt to $\|\cdot\|_2$-norm.

INPUT: $c = f_j(x) - f_m(x)$ and $v = \nabla f_m(x) - \nabla f_j(x)$ (j, desired class, m original class)
sort $\gamma_r = \max\{\frac{x_r-1}{v_r}, \frac{x_r}{v_r}\}$ in increasing order $\pi$
s=0; $\rho = 0$
**while** $\rho > c$ **do**
   $s \leftarrow s + 1$
   compute $\rho = \langle v, \delta(\gamma_{\pi_s}) \rangle$, where $\delta(\lambda)$ is the function defined in Lemma 4.1
**end while**
**if** $\rho \le c$ **then**
   compute $I_m = \{r \mid -x_r \le -\gamma_{\pi_{s-1})}v_r \le 1 - x_r\}$,
   compute $I_u = \{r \mid -\gamma_{\pi_{s-1}}v_r > 1 - x_r\}$,
   compute $I_l = \{r \mid -\gamma_{\pi_{s-1}}v_r < -x_r\}$
   $\lambda = \dfrac{\sum\limits_{r \in I_u} v_r(1-x_r) - \sum\limits_{r \in I_l} v_r x_r - c}{\sum\limits_{r \in I_m} v_r^2}$.
   $\delta = \max\{-x_r, \min\{-\lambda v_r, 1 - x_r\}\}$
**else**
   Problem has no feasible solution
**end if**

---

**Algorithm 2** Computation of box-constrained adversarial samples wrt to $\|\cdot\|_1$-norm.

INPUT: $c = f_j(x) - f_m(x)$ and $v = \nabla f_m(x) - \nabla f_j(x)$ (j, desired class, m original class)
sort $|v_i|$ in decreasing order
s:=0, g:=0,
**while** $g > c$ **do**
   $s \leftarrow s + 1$
   **if** $v_{\pi_s} > 0$ **then**
     $\delta_{\pi_s} = -x_{\pi_s}$
   **else**
     $\delta_{\pi_s} = 1 - x_{\pi_s}$
   **end if**
   $g \leftarrow g + \delta_{\pi_s} v_{\pi_s}$
**end while**
**if** $g \le c$ **then**
   $g \leftarrow g - v_{\pi_s}\delta_{\pi_s}$
   $\delta_{\pi_s} \leftarrow \frac{c-g}{v_{\pi_s}}$
**else**
   Problem has no feasible solution
**end if**

The Lagrangian of this problem is

$$L(\delta, t, \alpha, \beta, \gamma, \theta, \kappa, \lambda) = \langle t, \mathbf{1} \rangle + \lambda(\langle v, \delta \rangle - c) - \langle \alpha, \delta + x \rangle + \langle \beta, \delta - \mathbf{1} + x \rangle - \langle \gamma, \delta + t \rangle + \langle \theta, \delta - t \rangle - \langle \kappa, t \rangle$$
$$= \langle t, \mathbf{1} - \gamma - \theta - \kappa \rangle + \langle \delta, \beta - \alpha + \lambda v + \theta - \gamma \rangle + \langle \beta - \alpha, x \rangle - \langle \beta, \mathbf{1} \rangle - \lambda c$$

Minimization of the Lagrangian over $t$ resp. $\delta$ leads only to a non-trivial result if

$$\mathbf{1} - \gamma - \theta - \kappa = 0$$
$$\beta - \alpha + \lambda v + \theta - \gamma = 0$$

We get the dual problem

$$\max_{\alpha, \beta, \theta, \gamma, \kappa, \lambda} \ \langle \beta - \alpha, x \rangle - \langle \beta, \mathbf{1} \rangle - \lambda c \tag{12}$$
$$\text{sbj. to: } \mathbf{1} - \gamma - \theta - \kappa = 0$$
$$\beta - \alpha + \lambda v + \theta - \gamma = 0$$
$$\alpha \geq 0, \ \beta \geq 0, \ \theta \geq 0, \ \gamma \geq 0, \ \kappa \geq 0, \ \lambda \geq 0 \tag{13}$$

Using the equalities we can now simplify the problem by replacing $\alpha$ and using the fact that $\kappa$ is not part of the objective, the positivity just induces an additional constraint. We get

$$\alpha = \beta + \lambda v + \theta - \gamma.$$

Plugging this into the problem (16) we get

$$\max_{\beta, \theta, \gamma, \lambda} \ - \langle \lambda v + \theta - \gamma, x \rangle - \langle \beta, \mathbf{1} \rangle - \lambda c \tag{14}$$
$$\text{sbj. to: } \mathbf{1} - \gamma - \theta \geq 0$$
$$\beta + \lambda v + \theta - \gamma \geq 0$$
$$\beta \geq 0, \ \theta \geq 0, \ \gamma \geq 0, \ \lambda \geq 0 \tag{15}$$

We get the constraint $\beta \geq \max\{0, -\lambda v - \theta + \gamma\}$ (all the inequalities and functions are taken here componentwise) and thus we can explicitly maximize over $\beta$

$$\max_{\theta, \gamma, \lambda} \ - \langle \lambda v + \theta - \gamma, x \rangle - \sum_{i=1}^{d} \max\{0, -\lambda v_i - \theta_i + \gamma_i\} - \lambda c \tag{16}$$
$$\text{sbj. to: } \gamma + \theta \leq \mathbf{1}$$
$$\theta \geq 0, \ \gamma \geq 0, \ \lambda \geq 0 \tag{17}$$

As $0 \leq x_i \leq 1$ it holds for all $\theta \geq 0, \gamma \geq 0$

$$\langle -\lambda v - \theta + \gamma, x \rangle - \sum_{i=1}^{d} \max\{0, -\lambda v_i - \theta_i + \gamma_i\} \leq 0.$$

The maximum is attained if $\gamma_i - \theta_i - \lambda v_i = 0$ resp. with the constraints on $\gamma_i, \theta_i$ this is equivalent to $-1 \leq \lambda v_i \leq 1$. Suppose that $\lambda v_i > 1$ then the maximum is attained for $\gamma_i = 1$ and $\theta_i = 0$, and for $\lambda v_i < -1$ the maximum is attained for $\gamma_i = 0$ and $\theta_i = 1$. Thus by solving explicitly for $\theta$ and $\gamma$ we obtain

$$\max_{\lambda} \ \sum_{\lambda v_i > 1} (1 - \lambda_i v_i) x_i + \sum_{\lambda v_i < -1} (1 - \lambda_i v)(x_i - 1) - \lambda c \tag{18}$$
$$\text{sbj. to: } \lambda \geq 0 \tag{19}$$

Note that the first two terms are decreasing with $\lambda$ and the last term is increasing with $\lambda$. Let $\lambda^*$ be the optimum, then we have the following characterization

$$-1 < \lambda v_i < 1 \quad \Longrightarrow \quad \gamma_i + \theta_i < 1 \quad \Longrightarrow \kappa_i > 0 \quad \Longrightarrow t_i = 0 \Longrightarrow \delta_i = 0$$
$$\lambda v_i > 1 \quad \Longrightarrow \quad \gamma_i = 1, \theta_i = 0, \beta_i = 0, \alpha_i > 0 \quad \Longrightarrow \quad \delta_i = -x_i,$$
$$\lambda v_i < -1 \quad \Longrightarrow \quad \gamma_i = 0, \theta_i = 1, \beta_i > 0 \quad \Longrightarrow \quad \delta_i = 1 - x_i,$$

The cases $|\lambda v_i| = 1$ are undetermined but given that $\lambda > 0$ the remaining values can be fixed by solving for $c = \langle v, \delta \rangle$. The time complexity is again determined by the initial sorting step of $O(d \log d)$. The following linear scan requires $O(d)$. $\qquad \square$

Finally, we consider the case $p = \infty$.

$$\min_{\delta \in \mathbb{R}^d} \|\delta\|_\infty \tag{20}$$
$$\text{sbj. to: } f_j(x) - f_c(x) \geq \langle \nabla f_c(x) - \nabla f_j(x), \delta \rangle$$
$$0 \leq x_j + \delta_j \leq 1$$

**Lemma 4.3.** *Let $m = \arg\max_j f_j(x)$ and define $v = \nabla f_m(x) - \nabla f_j(x)$ and $c = f_j(x) - f_m(x) < 0$, then the solution of* (20) *can be found by solving for $t \geq 0$,*

$$\sum_{v_i > 0} v_i \max\{-t, -x_r\} + \sum_{v_i < 0} v_i \min\{t, 1 - x_r\} = c.$$

*which is done in Algorithm 3.*

*Proof.* We can rewrite the optimization problem into a linear program

$$\min_{t \in \mathbb{R}, \delta \in \mathbb{R}^d} t \tag{21}$$
$$\text{sbj. to: } c \geq \langle v, \delta \rangle$$
$$-x_j \leq \delta_j \leq 1 - x_j, \quad j = 1, \dots, d$$
$$-t \leq \delta_j \leq t, \quad j = 1, \dots, d$$
$$t \geq 0 \tag{22}$$

Thus we have $\max\{-t, -x_r\} \leq \delta_r \leq \min\{t, 1 - x_r\}$ for $r = 1, \dots, d$. Then it holds

$$\langle v, \delta \rangle \geq \sum_{v_r > 0} v_r \max\{-t, -x_r\} + \sum_{v_r < 0} v_r \min\{t, 1 - x_r\}$$

$$= -\sum_{v_r > 0, t \geq x_r} v_r x_r - t \sum_{v_r > 0, t < x_r} v_r + t \sum_{v_r < 0, t < 1 - x_r} v_r + \sum_{v_r < 0, t \geq 1 - x_r} v_r (1 - x_r),$$

and the lower bound can be attained if $\delta_r = \min\{t, 1 - x_r\}$ for $v_r < 0$ and $\delta_r = \max\{-t, -x_r\}$ for $v_r > 0$, $\delta_r = 0$ if $v_r = 0$. Note that both terms are monotonically decreasing with $t$. The algorithm 3 has complexity $O(d \log d)$ due to the initial sorting step followed by steps of complexity $O(d)$. $\qquad \square$

For nonlinear classifiers a change of the decision is not guaranteed and thus we use later on a binary search with a variable $c$ instead of $f_c(x) - f_j(x)$.

## 5 Experiments

The goal of the experiments is the evaluation of the robustness of the resulting classifiers and not necessarily state-of-the-art results in terms of test error. In all cases we compute the robustness guarantees from Theorem 2.1 (lower bound on the norm of the minimal change required to change the classifier decision), where we optimize over $R$ using binary search, and adversarial samples with the algorithm for the 2-norm from Section 4 (upper bound on the norm of the minimal change required to change the classifier decision), where we do a binary search in the classifier output difference in order to find a point on the decision boundary. Additional experiments can be found in the supplementary material.

**Kernel methods:** We optimize the cross-entropy loss once with the standard regularization (Kernel-LogReg) and with Cross-Lipschitz regularization (Kernel-CL). Both are convex optimization problems and we use L-BFGS to solve them. We use the Gaussian kernel $k(x, y) = e^{-\gamma \|x-y\|^2}$ where $\gamma = \frac{\alpha}{\rho_{\text{KNN40}}^2}$ and $\rho_{\text{KNN40}}$ is the mean of the 40 nearest neighbor distances on the training set and $\alpha \in \{0.5, 1, 2, 4\}$. We show the results for MNIST (60000 training and 10000 test samples). However, we have checked that parameter selection using a subset of 50000 images from the training set and evaluating on the rest yields indeed the parameters which give the best test errors when trained on the full set. The

---

**Algorithm 3** Computation of box-constrained adversarial samples wrt to $\|\cdot\|_\infty$-norm.

---

INPUT: $c = f_j(x) - f_m(x)$ and $v = \nabla f_m(x) - \nabla f_j(x)$ (j, desired class, m original class)
$d_+ := |\{l \,|\, v_l > 0\}|$ and $d_- := |\{l \,|\, v_l < 0\}|$
sort $\{x_l \,|\, v_l > 0\}$ in increasing order $\pi$, sort $\{1 - x_l \,|\, v_l < 0\}$ in increasing order $\rho$
s:=1, r:=1, g:=0, t:=0, $\kappa_+ = \sum_{v_l > 0} v_l$, $\kappa_- = \sum_{v_l < 0} v_l$, $\gamma_+ = \gamma_- = 0$.
**while** $g > c$ AND $(s \leq d_+$ OR $t \leq d_-)$ **do**
  **if** $x_{\pi_s} < 1 - x_{\rho_r}$ **then**
    $t = x_{\pi_s}, \quad \kappa_+ \leftarrow \kappa_+ - v_{\pi_s}, \quad \gamma_+ \leftarrow \gamma_+ - v_{\pi_s} x_{\pi_s},$
    $s \leftarrow s + 1$
  **else**
    $t = 1 - x_{\rho_r}, \quad \kappa_- \leftarrow \kappa_- - v_{\rho_r}, \quad \gamma_- \leftarrow \gamma_- - v_{\rho_r}(1 - x_{\rho_r}),$
    $t \leftarrow t + 1$
  **end if**
  $g = \gamma_+ + \gamma_- + t\,(\kappa_+ - \kappa_-)$
**end while**
**if** $g \leq c$ **then**
  undo last step
  $t = (c - \gamma_+ - \gamma_-)/(\kappa_- - \kappa_+)$
  compute $\delta_r = \begin{cases} \min\{t, 1 - x_r\} & \text{for } v_r > 0 \\ \max\{-t, -x_r\} & \text{for } v_r < 0 \end{cases}$
**else**
  Problem has no feasible solution
**end if**

---

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

For MNIST (all settings) the learning rate is for all methods chosen from $\{0.2, 0.5\}$. The regularization parameters for weight decay are chosen from $\{10^{-5}, 10^{-4}, 10^{-3}, 10^{-2}\}$, for Cross-Lipschitz from $\{10^{-5}, 10^{-4}, 5 * 10^{-4}, 10^{-3}\}$ and the dropout probabilities are taken from $\{0.4, 0.5, 0.6, 0.7\}$. For CIFAR10 the learning rate is for all methods chosen from $\{0.04, 0.1\}$, the regularization parameters for weight decay and Cross-Lipschitz are $\{10^{-5}, 10^{-4}, 5 * 10^{-4}, 10^{-3}\}$ and dropout probabilities are taken from $\{0.5, 0.6, 0.7, 0.8\}$. For CIFAR10 with data augmentation we choose the learning rate for all methods from $\{0.04, 0.1\}$, the regularization parameters for weight decay are $\{10^{-6}, 10^{-5}, 10^{-4}, 10^{-3}\}$ and

Adversarial Resistance (Upper Bound)
wrt to $L_2$-norm

Robustness Guarantee (Lower Bound)
wrt to $L_2$-norm

**Figure 2: Neural Networks, Left:** Adversarial resistance wrt to $L_2$-norm on MNIST. **Right:** Average robustness guarantee wrt to $L_2$-norm on MNIST for different neural networks (one hidden layer, 1024 HU) and hyperparameters. The Cross-Lipschitz regularization leads to better robustness with similar or better prediction performance. **Top row:** plain MNIST, **Middle:** Data Augmentation, **Bottom:** Adv. Training

for Cross-Lipschitz $\{10^{-5}, 10^{-4}, 5 * 10^{-4}, 10^{-3}\}$ and the dropout probabilities are taken from $\{0.5, 0.6, 0.7, 0.8\}$. Data augmentation for MNIST means that we apply random rotations in the angle $[-\frac{\pi}{20}, \frac{\pi}{20}]$ and random crop from 28x28 to 24x24. For CIFAR-10 we apply the same and additionally we mirror the image (left to right) with probability 0.5 and apply random brightness $[-0.1, 0.1]$ and random contrast change $[0.6, 1.4]$. In each substep we ensure that we get an image in $[0, 1]^d$ by clipping. We implemented adversarial training by generating adversarial samples wrt to the infinity norm with the code from Section 4 and replaced 50% of each batch as adversarial samples. Finally, we use for SGD batchsize 64 in all experiments.

**Illustration of adversarial samples:** we take one test image from MNIST and apply the adversarial generation from Section 4 wrt to the 2-norm to generate the adversarial samples for the different kernel methods and neural networks (plain setting), where we use for each method the parameters leading to best test performance. All classifiers change their originally correct decision to a "wrong" one. It is interesting to note that for Cross-Lipschitz regularization (both kernel method and neural network) the "adversarial" sample is really at the decision boundary between 1 and 8 (as predicted) and thus the new decision is actually correct. This effect is strongest for our Kernel-CL, which also requires the strongest modification to generate the adversarial sample. The situation is different for neural networks, where the classifiers obtained from the two standard regularization techniques are still vulnerable, as the adversarial sample is still clearly a 1 for dropout and weight decay. We show further examples below.

**Figure 3:** **Left:** Adversarial resistance wrt to $L_2$-norm on test set of CIFAR10. **Right:** Average robustness guarantee on the test set wrt to $L_2$-norm for the test set of CIFAR10 for different neural networks (one hidden layer, 1024 HU) and hyperparameters. While Cross-Lipschitz regularization yields good test errors, the guarantees are not necessarily stronger. **Top row:** CIFAR10 (plain), **Middle:** CIFAR10 trained with data augmentation, **Bottom:** Adversarial Training.

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

**Figure 11:** Top left: original test image, for each classifier we generate the corresponding adversarial sample which changes the classifier decision (denoted as Pred). Note that for the kernel methods this new decision makes sense, whereas for all neural network models the change is so small that the new decision is clearly wrong.

**Figure 13:** **Left:** Adversarial resistance wrt to $L_2$-norm on test set of the german traffic sign benchmark (GTSB) in the plain setting. **Right:** Average robustness guarantee on the test set wrt to $L_2$-norm for the test set of GTSB for different neural networks (one hidden layer, 1024 HU) and hyperparameters. Here dropout performs very well both in terms of performance and robustness.

| Original, Class 5 | K-SVM, Pred:9, $\|\delta\|_2 = 1.5$ | K-CL, Pred:9, $\|\delta\|_2 = 2.2$ |
| NN-WD, Pred:9, $\|\delta\|_2 = 1.1$ | NN-DO, Pred:9, $\|\delta\|_2 = 1.0$ | NN-CL, Pred:9, $\|\delta\|_2 = 1.4$ |

**Figure 12:** Top left: original test image, for each classifier we generate the corresponding adversarial sample which changes the classifier decision (denoted as Pred). Note that for the kernel methods this new decision makes sense, whereas for all neural network models the change is so small that the new decision is clearly wrong.

**German Traffic Sign Benchmark:**  As a third dataset we used the German Traffic Sign Benchmark (GTSB) [23], which consists of images of german traffic signs, which has 43 classes with 34209 training and 12630 test samples. The results are shown in Figure 13. For this dataset Cross-Lipschitz regularization improves the upper bounds compared to weight decay but dropout achieves significantly better prediction performance and has similar upper bounds. The robustness guarantees for weight decay and Cross-Lipschitz are slightly better than for dropout.

**Residual Networks:**  All experiments so far were done with one hidden layer neural networks so that we can evaluate lower and upper bounds. Now we want to demonstrate that Cross-Lipschitz regularization can also successfully be used for deep networks. We use residual networks proposed in [9] with 32 parameter layers and non-bottleneck residual blocks. We follow basically their setting, apart from that we did not subtract the per-pixel mean so that all images are in $[0, 1]^d$ and use random crop but without any padding as in [9]. Similar to [9], we train for 160 epochs, and the learning rate is divided by 10 on the 115-th and 140-th epochs. For the experiments with dropout we followed the recommendation of [25], inserting a dropout layer between convolutional layers inside each residual block. For Cross-Lipschitz regularization we use automatic differentiation in TensorFlow [1] to calculate the derivative with respect to the input, which slows done the training by a factor of 10.

For the plain setting the learning rate for all methods is chosen from $\{0.2, 0.5\}$, except for the runs without regularization, for which it is from $\{0.08, 0.1, 0.2, 0.4, 0.6, 0.8\}$. For weight decay the regularization parameter is chosen from $\{10^{-5}, 10^{-4}, 10^{-3}, 10^{-2}\}$, for Cross-Lipschitz from $\{10^{-4}, 10^{-3}, 10^{-2}, 10^{-1}\}$, and for dropout the probabilities are from $\{0.5, 0.6, 0.7, 0.8\}$. For the data augmentation setting the only difference was in the higher learning rates: no

Adversarial Resistance (Upper Bound)
wrt to $L_2$-norm (ResNets)

Adversarial Resistance (Upper Bound)
wrt to $L_2$-norm (ResNets)

**Figure 14:** Results on CIFAR10 for a residual network with different regularizers. As we only have lower bounds for one hidden layer networks, we can only show upper bounds for adversarial resistance. **Left:** with data augmentation similar to [25] **Right:** plain setting

regularization - $\{0.2, 0.5, 0.8, 1.0, 1.5, 2.0, 3.0, 4.0\}$, weight decay - $\{0.1, 0.4\}$, Cross-Lipschitz - $\{0.2, 1.0\}$. The results are shown in Figure 14. Cross-Lipschitz regularization improves the upper bounds on the robustness against adversarial manipulation compare to weight decay and dropout by a factor of 2 to 3 both in the plain setting (right) and with data augmentation (left). This comes at a price of a slightly worse test performance. However, it shows that Cross-Lipschitz regularization is also effective for deep neural networks. It remains interesting future work to come up also with interesting instance-specific lower bounds (robustness guarantees) for deep neural networks.

**Outlook** Formal guarantees on machine learning systems are becoming increasingly more important as they are used in safety-critical systems. We think that there should be more research on robustness guarantees (lower bounds), whereas current research is focused on new attacks (upper bounds). We have argued that our instance-specific guarantees using our local Cross-Lipschitz constant is more effective than using a global one and leads to lower bounds which are up to 8 times better. A major open problem is to come up with tight lower bounds for deep networks.

## Footnotes

[1] The Lipschitz constant $L$ wrt to $p$-norm of a piecewise continuously differentiable function is given as $L = \sup_{x \in \mathbb{R}^d} \|\nabla f(x)\|_q$. Then it holds, $|f(x) - f(y)| \leq L \|x - y\|_p$.

[2]Note that then the optimization of $R$ in Theorem 2.1 would be unnecessary.