[Reviews · NeurIPS 2017]

Reviewer 1



This paper proves that under certain assumptions one needs to change an instance substantially so that the value of the classifier changes. The goal is to understand the effects of adversarial manipulation of various inputs. Prior work had shown examples where a tiny adversarial change can change the value of the input, i.e., a total lack of stability of the classifier.

Reviewer 2



This paper fills an important gap in the literature of robustness of classifiers to adversarial examples by proposing the first (to the best of my knowledge) formal guarantee (at an example level) on the robustness of a given classifier to adversarial examples. Unsurprisingly, the bound involves the Lipschitz constant of the Jacobians which the authors exploit to propose a cross-Lipschitz regularization. Overall the paper is well written, and the material is well presented. The proof of Theorem 2.1 is correct. I did not check the proofs of the propositions 2.1 and 4.1. This is an interesting work. The theorem 2.1 is a significant contribution. The experimental section is not very insightful as very simple models are used on small/toy-ish datasets. Some claims in the paper are not consistent: - "the current main techniques to generate adversarial samples [3, 6] do not integrate box constraints" There exist approaches integrating box constraints: see [1,2] - "It is interesting to note that classical regularization functionals which penalize derivatives of the resulting classifier function are not typically used in deep learning." Penalizing the derivatives is a well-known strategy to regularize neural networks[3,4]. In fact, it has recently been exploited to improve the robustness of neural networks to adversarial examples[3]. I have the following question for the authors: Theorem 2.1 says that the classifier decision does not change inside a ball of small radius for a given p-norm. Yet, we can still find adversarial examples even when the norm of the gradient is small (yes, we can. I have tried!). Does this mean the topology induced by the p-norm does not do justice to the perceptual similarity we have of the image and its adversarial version? In other words; Does close in terms of p-norm reflect "visually close"? Overall I like the paper. Though it leaves me with mixed feelings due to weak experimental section, I lean on the accept side. [1] Intriguing properties of neural networks [2] DeepFool: a simple and accurate method to fool deep neural networks [3] Parseval Networks: Improving Robustness to Adversarial Examples [4] Double back propagation: increasing generalization performance